# Influence of Heat Treatment of Steel AISI316L Produced by the Selective Laser Melting Method on the Properties of Welded Joint

**DOI:** 10.3390/ma15051690

**Published:** 2022-02-24

**Authors:** Petr Mohyla, Jiri Hajnys, Lucie Gembalová, Andrea Zapletalová, Pavel Krpec

**Affiliations:** 1Department of Mechanical Technology, Faculty of Mechanical Engineering, Technical University of Ostrava, 708 00 Ostrava, Czech Republic; petr.mohyla@vsb.cz; 2Department of Machining, Faculty of Mechanical Engineering, Technical University of Ostrava, 708 00 Ostrava, Czech Republic; 3Department of Physics, Faculty of Electrical Engineering and Computer Science, Technical University of Ostrava, 708 00 Ostrava, Czech Republic; lucie.gembalova@vsb.cz; 4Institute of Languages, Technical University of Ostrava, 708 00 Ostrava, Czech Republic; andrea.zapletalova@vsb.cz; 5V-NASS, A.S., Halasova 2938/1a, 703 00 Ostrava, Czech Republic; pavel.krpec@v-nass.cz

**Keywords:** AISI 316L, heat treatment, tungsten inert gas welding, selective laser melting, mechanical properties, microstructural analysis

## Abstract

This work is focused on the influence of heat treatment of a part produced by the SLM (selective laser melting) method of stainless steel, 316L. Two heat treatment regimens were tested and compared with the state without heat treatment. Subsequently, TIG (tungsten inert gas) welds were created on the base materials processed in this way. All welds were subjected to mechanical tests and microstructural analysis. The tensile test was performed both for the welded joint and for the base material in the transverse and longitudinal directions. The tensile strength values of the samples with the welded joint were compared with the values required for the base material, 316L forged steel (1.4404). Microstructural analysis revealed significant differences between samples with and without heat treatment. The results of these tests are supported by SEM analysis. EDAX (energy dispersive analysis of X-rays) semiquantitative analysis confirmed the presence of ultra-fine pores in the structure. The results of mechanical tests show that the solution annealing at 1040 °C for 0.5 h gives better results than the same heat treatment with a duration of 2 h.

## 1. Introduction

The selective laser melting (SLM) process enables the production of strong, lightweight, and complex metal structures [1]. SLM is one of the most promising metal 3D printing technologies available today. It is commonly used for rapid prototyping but is already slowly reaching mass production, especially for smaller pieces and batches [2]. The range of available printable metal alloys for the SLM method is currently limited to a few selected materials, mainly to guarantee their weldability between vectors and layers. The SLM method commonly offers the production of stainless steel, tool steel, Inconel, chromium–cobalt, aluminum, and titanium alloys [3]. Recently, studies have begun to emerge that attempt to combine two different materials into one another to create a bimetallic material that combines and utilizes the properties of both materials. For example, Chen’s team made a successful attempt, which managed to find the right printing parameters and subsequently printed a multimaterial based on 316/CuSn10 using SLM [4].

SLM is a very energy-intensive process because each layer of metal powder must be heated above the melting point of the metal. High-temperature gradients that occur during SLM production can also lead to stresses and deformations within the final product [5,6,7,8]. Furthermore, the printing process can result in the sticking of partially melted powder or spattering of the melt, as well as deformation of the product, formation of a crack, or separation of the component from the base platform [9,10].

However, apart from the process shortcomings mentioned, additive manufacturing (AM) offers advantages such as weight optimization, complex geometries, material savings, environmental friendliness, etc. [11,12]. In terms of the final product, AM is limited by the size of the object dimensions. One of the solutions is to join two parts manufactured by AM together or to attach them to an existing structure [13]. For this reason, welding is in play, which is one of the most common methods of joining two metallic materials. The welding process itself has been known for several decades and is also well studied [14,15,16], however, welding in connection with additively manufactured parts has been investigated very little and there are only a few studies that address this topic [17,18,19,20].

The aforementioned ailments were the motivation for the creation of this publication, where the authors focus on additively manufactured 316L stainless steel material in relation to the effect of welding and the influence of heat treatment on the mechanical properties and microstructure of the weld joint and the base material. The tungsten inert gas (TIG) welding method was used as the welding process. The welded joints were tested to determine the effect of heat treatment on the microstructure and mechanical properties. Hence, in this work, heat treatment is carried out in two different sets. Based on the results obtained, it is possible to find the best welding parameters to qualify the welding procedure (WPQR).

## 2. Experimental Methods

The fabrication and preparation of the test specimens included: the fabrication of the specimens using SLM technology, the TIG welding of the specimens, the preparation of the welded specimens for mechanical testing, mechanical testing, and microstructural analysis.

### 2.1. Powder Characterisation

The material used for the SLM was a 316L austenitic stainless-steel powder with particle size ranging from 15 to 45 μm [18]. The powder used was supplied by Renishaw (Wotton-under-Edge, U.K.) and was sieved several times. In the sieved powder, there is only a minimal change in particle morphology resulting in a slightly increased surface roughness; however, the chemical composition remains the same. Therefore, the powder can be reused successfully, as demonstrated by previous studies [21,22]. The chemical composition of the powder used is shown in Table 1, which was determined by glow discharge optical emission spectrometry (GDOES) using a spectrum analytic optical emission spectrometer in accordance with the work [23,24].

### 2.2. SLM Sample Production and Application of Heat Treatment

A total of six pieces of test plates (sheet metal) were made. The main dimensions of the test plates were: 150 mm × 135 mm × 10 mm (W × H × D). Plates were manufactured using the university’s Renishaw AM 400 Selective Laser Melting (SLM) machine (Wotton-under-Edge, U.K.). This machine uses a 400 W ytterbium fiber laser, which is integrated into the system control hardware and software. Argon with purity 5.0 was chosen as the inert shielding gas. All samples were printed in one structure with identical process parameters; see Table 2. After printing, samples were cut from the substrate, and heat treatment without substrate was applied. Figure 1 shows the fabricated plates (specimens).

In total, three test welds were prepared, marked A, B, and C, which differentiated the heat treatment after printing. Specimen A was left in as-built condition. Test samples B and C were heat treated by annealing in a LAC Ht 40AL furnace (LAC, Zidlochovice, Czech Republic) before welding. An overview of the heat treatment of the particular samples is given in Table 3. In the first stage of heat treatment, heating was carried out at a rate of 250 °C per hour to an annealing temperature of 1040 °C, with an annealing time of 0.5 h and 2 h, see Figure 2. This was followed by accelerated cooling with a fan for 30 min. This annealing solution is used to dissolve carbides. The set of temperature and an annealing time is in accordance with the publications [25,26]. The samples were heated in an oven under an inert argon atmosphere. The argon supply was 8 L/min.

### 2.3. Welding of Specimens

The TIG method (141) was used to weld all test specimens. Welding was performed without filler material and the plates were assembled without any gaps. The thickness of the welded plates at the joint (butt joint) was 2.5 mm, other dimensions of the welded part are shown in Figure 3. The welding current ranged from 80–90 A and the welding voltage ranged from 10–12 V. The welding speed ranged from 1.8 to 2.0 mm/s and the heat input ranged from 0.26 to 0.36 kJ/mm. Argon shielding gas was used both for weld and root protection. The gas flow rate for weld protection was 12 L/min and 6 L/min for root protection.

### 2.4. Tensile Tests

To determine the tensile test, specimens were cut from the welded plates. Figure 4 shows graphically in which position and from which location the specimens were cut. The specimens placed only in the base material (without weld) are circular in cross section and were placed both in the direction of the structure, that is, vertically (across the material layers) and horizontally (along the material layers). The samples across the weld joint are always placed vertically (across printing layers) and are rectangular in cross section. The samples were tested on a Zwick/Roell Z250 machine or a Zwick/Roell Z600 (Zwick Roell Group, Ulm, Germany). Testing parameters were set to: strain rate up to yield strength set to 0.00025 s^−1^ and strain rate after yield point and up to yield strength set to 0.0067 s^−1^.

### 2.5. Hardness Test

The hardness test of the welded joint was performed on all welds. Indentation started from the base material, continued through the heat-affected zone (HAZ), and then the weld metal to the HAZ and base material on the other side of the weld. The location of the indenters can be seen in Figure 5. The hardness was measured using the Vickers method with a load of HV10. Hardness properties were measured on a WPM Leipzig 300/436 hardness tester (WPM GmbH, Leipzig, Germany). In each case, 19 indentations were made across the entire welded joint and base material. Due to the small thickness of the sample (2.5 mm), the line of indenters was situated approximately in the middle of the sample.

### 2.6. Optical Microscope Measurements

For microstructural analysis, the samples were prepared in a standard way, i.e., by grinding, polishing, and subsequent electrochemical etching. An Olympus GX- 51 metallographic microscope (Olympus, Tokyo, Japan) was used to observe the microstructures.

### 2.7. SEM (Scanning Electron Microscope) Analysis

The sample with heat treatment C (1040 °C/2 h) was subjected to SEM analysis. The analysis was performed on a device: scanning electron microscope FEI Quanta 650 FEG (Thermo Fisher Scientific, Hillsboro, OR, USA); equipment settings: high voltage: 20 kV; detector: BSED (backscattered electron detector); and vacuum pressure: 50 Pa.

## 3. Results

### 3.1. Evaluation of Tensile Test in Base Material

Tensile tests were performed on standard test specimens of a circular cross section. The results were compared with the standard EN 10028-7 [27] “Flat products made of steels for pressure purposes–Part 7: Stainless steels”. According to this standard, the required ultimate strength must be between 520 and 670 MPa, the yield strength a minimum of 220 Mpa, and the ductility in the transverse direction a minimum of 45%. Individual results are shown in Figure 6. The green lines represent the minimum and maximum according to EN 10028-7 [27] and the red lines represent the typical mechanical properties declared by the 316L powder manufacturer Renishaw (676 MPa in the horizontal direction and 624 MPa in the vertical direction). The test results showed that specimens made horizontal in the direction of construction have approximately 70 to 200 MPa higher strength than vertical specimens.

The minimum required elongation according to the standard cited [27] is shown in Figure 7 by the green line (45%) and the red lines represent the typical ductility values declared by the powder manufacturer (43% in the horizontal direction and 35% in the vertical direction). The horizontal specimen with a heat treatment of setting C (1040 °C/2 h) was the closest to the required value. The same trend was spotted in the vertical specimens and direction.

### 3.2. Evaluation of Tensile Test in Welded Joint

Flat test specimens were fabricated for the transverse tensile test of the welded joint. Then, they were subjected to tensile testing on the Zwick/Roell Z600 machine. The results obtained are shown in Figure 8.

From the results obtained, it is evident that the ultimate strength values were the highest for the welded joint without heat treatment followed by the specimen annealed for 0.5 h. The green lines on the graph show the minimum and maximum strength according to EN 10028-7 [27] and the red lines represent the typical mechanical properties declared by the 316L powder manufacturer Renishaw (624 MPa), as shown in Figure 8.

### 3.3. Evaluation of Hardness Test

Figure 9 provides a comparison of the hardness of the specimens without heat treatment with the heat-treated ones. The figure shows that the weld joint without heat treatment exhibits a higher hardness than the annealed specimens, but only in the region of the base material.

### 3.4. Evaluation of Microstructure

The typical microstructures of the base material are shown in Figure 10. In annealed material, a pure austenitic structure occurs. After heat treatment at 1040 °C, the typical structure that occurs after 3D printing of the powdered metal cannot be observed. The typical structure consists of solute bands that resemble weld beads; see Figure 10a. However, the distribution of the “black particles” of the sample in Figure 10b indicates the shape of the original solute bands.

Figure 10c,d shows a comparison of the microstructures around the fusion boundary. In both cases, the original 3D printing structure was completely dissolved on the side of the base material. The HAZ microstructure in both cases is pure austenitic with a minimum of carbides or pores. Heat treatment (Figure 10d) caused only a coarsening of the austenitic grain in the HAZ. The microstructure of the weld metal is shown in Figure 10e,f. In the structure, we observe austenite and delta ferrite, which are excluded interdendritically, so it is a dendritic structure. The difference between the sample on Figure 10e (without heat treatment) and sample 10f (heat treatment 1040 °C/2 h) is that the weld metal structure is finer than that of sample 10f.

### 3.5. Evaluation of SEM Analysis

The aim of SEM analysis was to determine the composition of the “black particles” that occur in the place of solute banding lines. Figure 11 shows typical distribution of “black particles” in the austenitic matrix. Figure 12 shows two analyzed particles marked by arrows. The initial assumption was that the “black particles” were precipitates such as chromium carbides. For better understanding, EDAX (energy dispersive analysis of X-rays) semiquantitative chemical composition analysis was performed on the areas of “black particles”. The results of this analysis are shown in Figure 13 and Table 4. The chemical composition corresponds to the composition of the base material, steel 316L, in terms of Mo, Cr, and Ni. The content of carbon is not relevant to the EDAX method. Thus, these “particles” are most likely pores, not precipitates. We obtained the same results for all measurements on these “black particles”.

## 4. Discussion

### 4.1. Mechanical Properties Measurement

The results of the tensile tests in the base material showed that the specimens that are made in the horizontal direction have approximately 70 to 200 MPa higher strength than the specimens in vertical direction. All horizontal-oriented specimens meet the strength requirements of the standard EN 10028-7 with a margin. Ultimate strength values in the vertical direction were the highest for the welded joint without heat treatment followed by the specimen annealed at 1040 °C for 0.5 h. Only these two samples (in the vertical direction) meet the requirements of the standard EN 10028-7 [23]. The specimen heat treated at 1040 °C for 2 h has the lowest strength values in the vertical direction. The elongation of the samples in the base material was insufficient compared with EN 10028-7 in all cases. Low elongation values are typical for 3D prints made of 316L steel; similar conclusions were reached by the authors in [10].

From the results obtained in the welded joint, it can be seen that the ultimate strength values were highest in the welded joint without heat treatment followed by the 0.5 h/1040 °C annealed specimen. Only these two specimens meet the requirements of the standard 10028-7 for the minimum strength value. The low values for the transverse test are due both to the orientation of the test bars being vertical printing layers and to the fact that the weld was made without any filler material, and therefore the strength of the joint could not be increased by a filler material with a higher strength value.

The results of the hardness measurement show that the welded joint without heat treatment shows a higher hardness than the annealed samples. The difference is evident especially in the base material, and hardness decreases with longer annealing time. The hardness in the weld metal is lower in all three cases compared with the base metal. This is because filler metal was not used during welding. In HAZ, hardness is affected by more influences (grain size, defects, and hard particles) and therefore the course of hardness is more random in this area.

### 4.2. Microstructure Measuerement

Microstructural analysis revealed a significant difference in the sample without heat treatment compared with the heat-treated samples. In the sample without heat treatment, the characteristic 3D printing structure in the form of solute bands is seen. The heat treatment has caused partial recrystallization, and the structure has changed to pure austenite. However, in the heat-treated samples, the appearance of “black particles” located along the solute banding lines was observed.

### 4.3. SEM Analysis

SEM analysis revealed the origin of the “black particles”. The results of the EDAX analysis show that the chemical composition of the investigated particles corresponds to the composition of the base material, 316L steel. The structure of the analyzed particles also corresponds to the pores. Therefore, we can state that it is most likely pores, not precipitates. These conclusions correspond to the results of other work [10], where it is stated that depending on the laser power and scanning speed, solute bands in SLM-fabricated 316L can be enriched or depleted.

## 5. Conclusions

This work is a contribution to the development of SLM technology, especially in the field of post-processing, which is an important part of additive technologies. The obtained results lead to clarification of the influence of welding and heat treatment on the mechanical properties and microstructure of the weld joint of 3D-printed steel, AISI316L. The following conclusions can be drawn from the obtained results:Microstructural analysis confirmed the existence of ultra-fine pores, copying the original solute banding lines, which arise in 3D printing.The ultimate tensile strength of the samples in vertical direction are significantly lower in comparison with samples in the horizontal direction, which makes a difference of up to 200 MPa.The elongation of the samples in the base material in all cases did not meet the requirements of the standard for cast and rolled steel, 316L.Solution annealing increases elongation of the base material, but elongation still does not enrich the values of hot-rolled steel.In terms of tensile strength, the solution annealing at 1040 °C for 0.5 h gives better results than the same heat treatment for 2 h.The TIG method is suitable for welding 3D-printed steel, 316L, however, the use of a suitable filler metal may be recommended to achieve a higher tensile strength and hardness in weld metal.

New experiments and measurements are currently underway to extend the results. The knowledge gained will lead to the development of heat treatment and welding of 3D-printed steel AISI316L.

## Figures and Tables

**Figure 1 materials-15-01690-f001:**
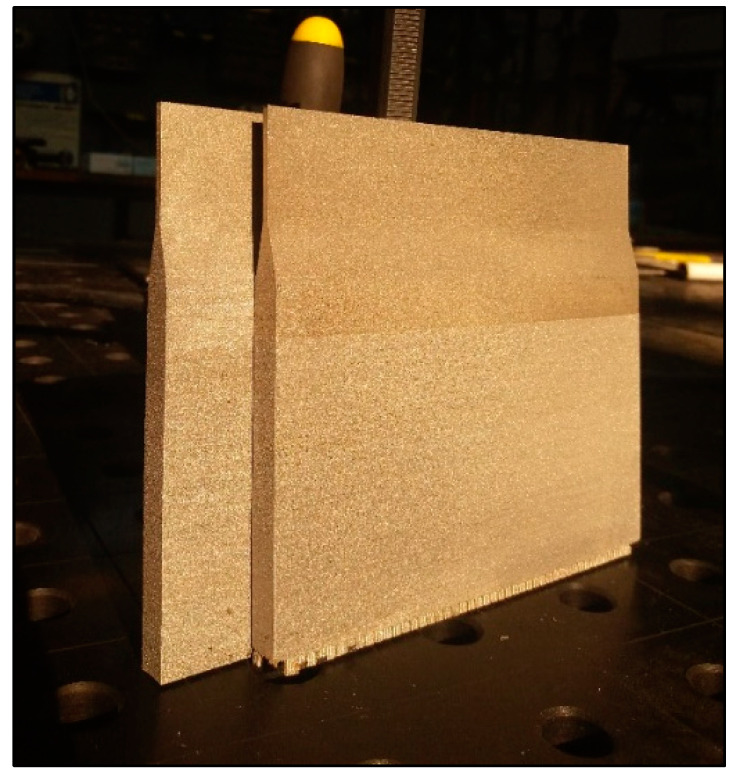
Test plates made by the SLM method ready for welding.

**Figure 2 materials-15-01690-f002:**
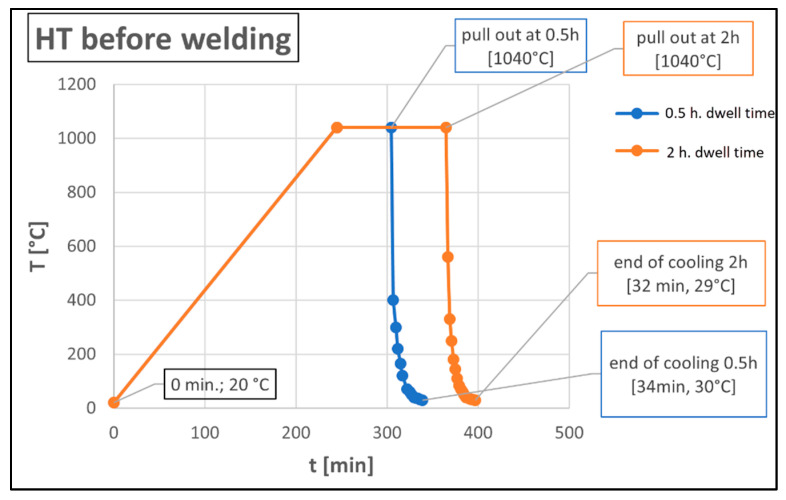
Diagram of individual heat treatment.

**Figure 3 materials-15-01690-f003:**
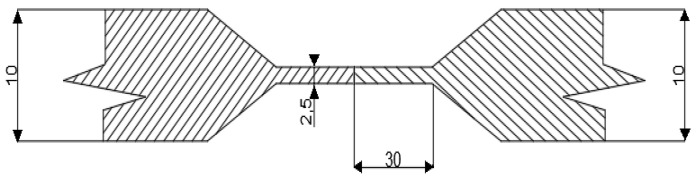
Schematic of the experimental welded joint arrangement (dimensions are in mm).

**Figure 4 materials-15-01690-f004:**
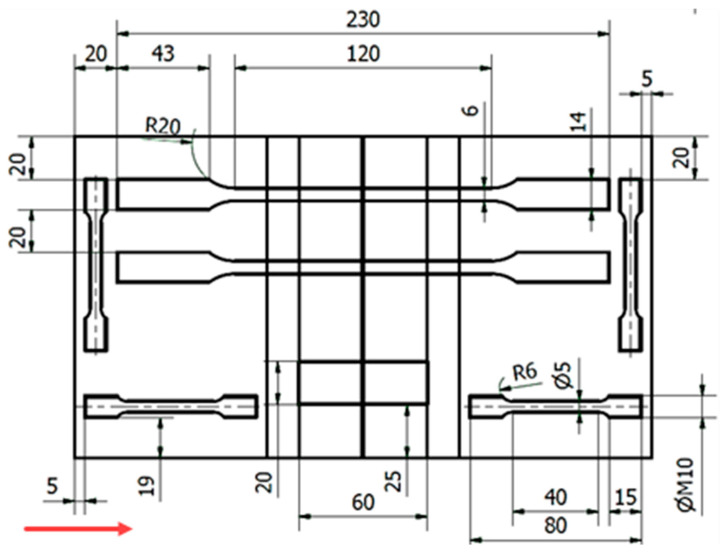
Location of test specimens for tensile test on welding plate (dimensions are in mm); red arrow indicates build direction.

**Figure 5 materials-15-01690-f005:**
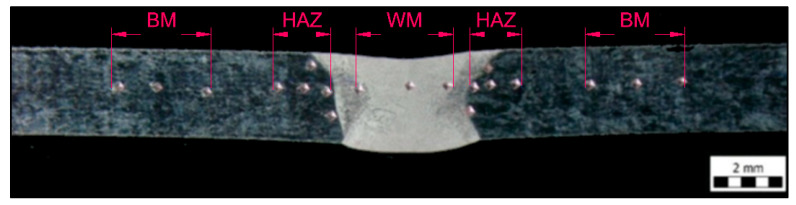
Layout of indenters during the hardness test, Vickers HV10.

**Figure 6 materials-15-01690-f006:**
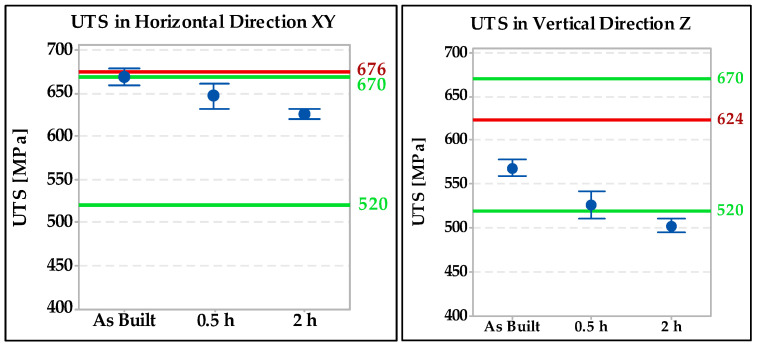
Ultimate tensile strength test results in the base material specimens.

**Figure 7 materials-15-01690-f007:**
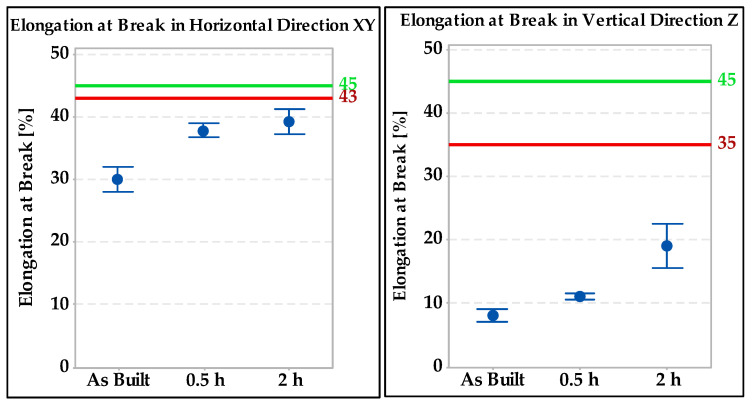
Comparison of elongation values in base material specimens.

**Figure 8 materials-15-01690-f008:**
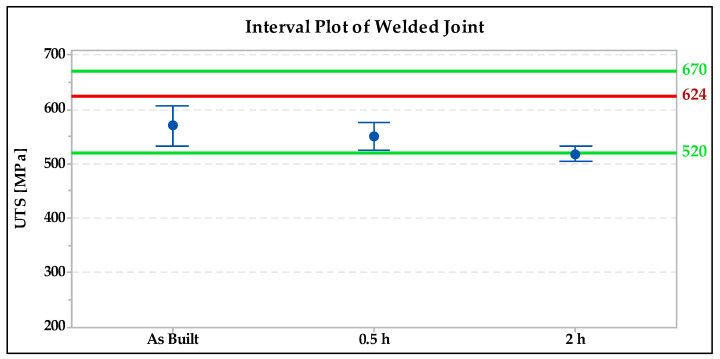
Interval plot of UTS (ultimate tensile stress) of transverse tensile test of welded joint.

**Figure 9 materials-15-01690-f009:**
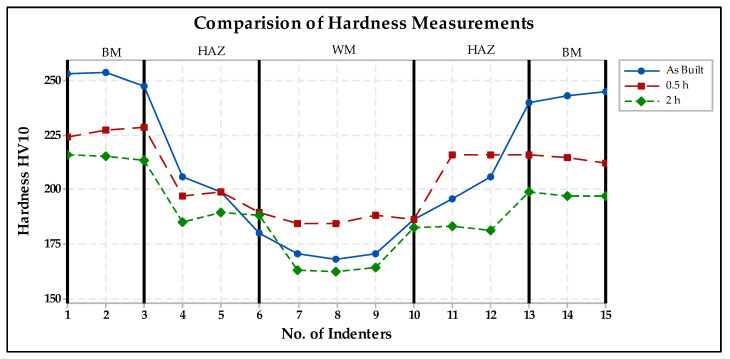
Comparison of hardness in welded joint.

**Figure 10 materials-15-01690-f010:**
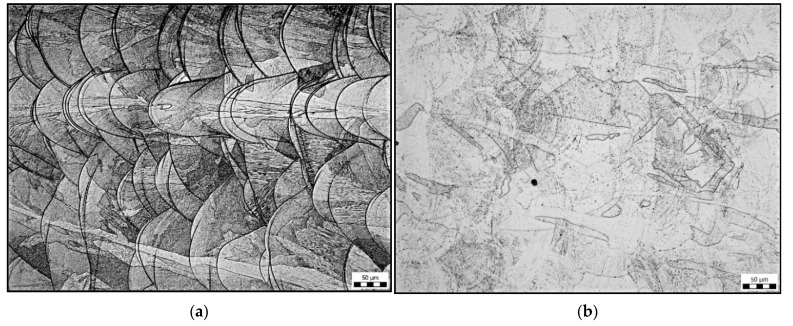
Microstructures in the base material: (**a**) without heat treatment; (**b**) 1040 °C/2 h, microstructure at the fusion boundary, left weld metal, right HAZ; (**c**) without heat treatment; (**d**) heat treatment 1040 °C/2 h, microstructure of weld metal; (**e**) without heat treatment; and (**f**) heat treatment 1040 °C/2 h.

**Figure 11 materials-15-01690-f011:**
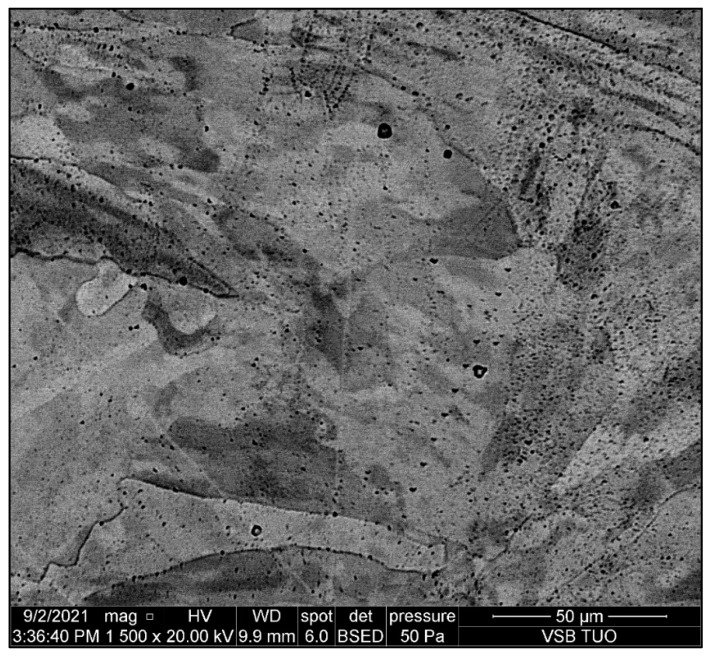
Microstructure of base metal and pores in matrix; magnification 1500×.

**Figure 12 materials-15-01690-f012:**
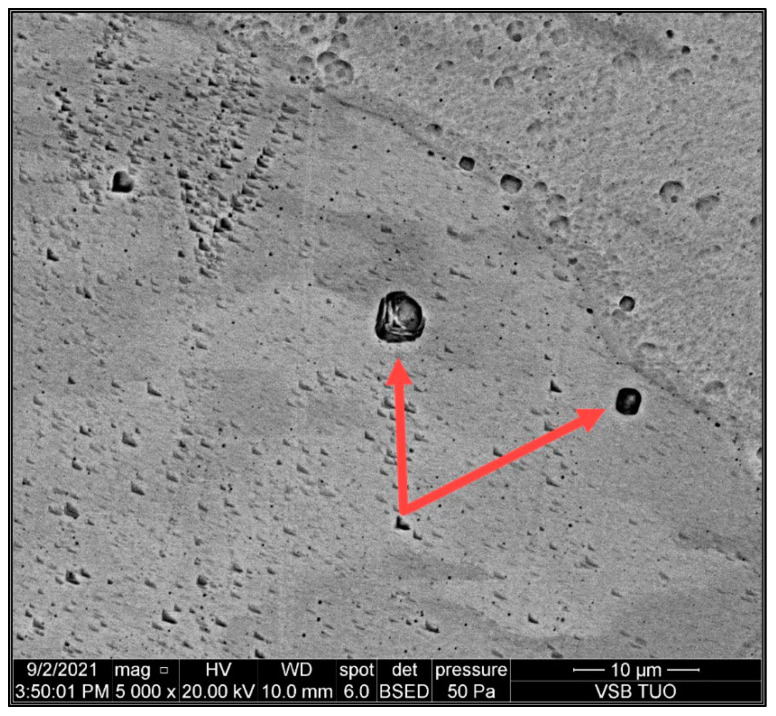
Analyzed pores in matrix (indicated by red arrows); magnification 5000×.

**Figure 13 materials-15-01690-f013:**
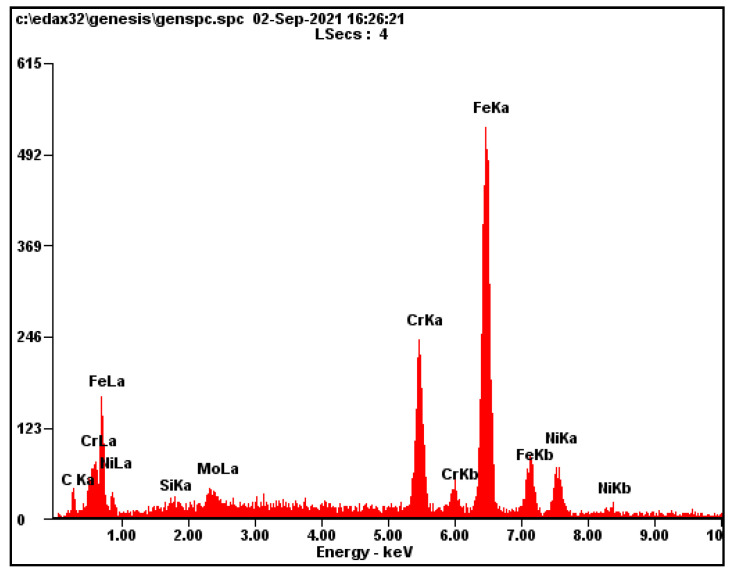
EDAX semiquantitative analysis of “black particles”.

**Table 1 materials-15-01690-t001:** Chemical composition of AISI 316L base material in wt.%.

C	Mn	Si	P	S	Cr	Ni	Mo	W	Cu	Ti	Nb	Al	B	Fe
0.016	1.17	0.22	0.023	0.0067	17.72	14.24	2.73	0.19	0.077	0.0003	0.013	0.01	0.002	balance

**Table 2 materials-15-01690-t002:** Settings of process parameters for production SLM specimens.

Laser Power (W)	Scanning Speed (mm/s)	Layer Thickness (µm)	Strategy	Focus Size (µm)	Melting Range (°C)
200	650	50	Chessboard	70	1371 to 1399

**Table 3 materials-15-01690-t003:** Overview of heat treatment of experimental samples.

Sample	Heat Treatment
A	-
B	1040 °C/0.5 h
C	1040 °C/2.0 h

**Table 4 materials-15-01690-t004:** Semiquantitative analysis in wt.%.

Si	Mo	Cr	Ni	Fe
0.67	2.48	16.33	12.54	balance

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
