# Peer review of "Influence of Heat Treatment of Steel AISI316L Produced by the Selective Laser Melting Method on the Properties of Welded Joint"

_materials, 2022, doi:10.3390/ma15051690_

Round 1

Reviewer 1 Report

Comments and Suggestions for Authors

This study investigates the effect of heat treatment on the microstructural and mechanical properties of the selective-laser-meted 316L weld joint. Overall, the manuscript lacks organization and the quality is low. I found some crucial drawbacks that need to be addressed. First of all, the novelty of the current manuscript is low. In fact, the results of the study only show that the heat treatment, which is basically solution annealing, attributes to the partial recrystallization of the stainless steel weld. The effect of solution annealing on stainless steel has been studied extensively in the past decades. There is nothing new in the current manuscript. Second, the discussion of the experimental results is insufficient and superficial. Not all of the testing results have been discussed. A systematic and comprehensive discussion is needed to enhance the scientific soundness of the study. Third, the characterization of the “black particles” is questionable. It is clear in the results of the chemical analysis that the carbon content of the “black particles” is much higher than the base metal, indicating that the “black particles” is actually chromium carbide. However, the authors claimed that these “black particles” just pores. There is also no explanation of how these pores were formed. Lastly, the manuscript contains many unclear statements, grammatical errors and incorrect punctuations, which make the readers difficult to understand. I strongly recommend the manuscript being polished by a native English speaker to make it more readable. Based on the above comments, I do not recommend publication of this manuscript. Some of my other comments are listed below:

Page 1: “Influence of Heat Treatment of Steel AISI316L Produced by the SLM Method on the Properties of Welded Joint

The abbreviation “SLM” should be spelled out in the title.

Page 1: “This work is focused on the influence of heat treatment of a part produced by SLM method of stainless steel 316L.

The abbreviation “SLM” should be spelled out during first appearance in the abstract.

Page 1: “TIG welds were created on the base materials processed in this way.

The abbreviation “TIG” should be spelled out during first appearance in the abstract.

Page 1: “Microstructural analysis revealed significant differences between samples with and without heat treatment. The results of these tests have been supported by SEM analysis. EDAX semiquantitative analysis confirmed the presence of ultra-fine pores in the structure.

The abbreviation “SEM” and “EDAX” should be spelled out during first appearance in the abstract. In addition, the important findings in the current study is not well indicated in the abstract. Please add more results of the current manuscript into the abstract.

Page 1: “Keywords: AISI316L; heat treatment; welded joint; SLM; mechanical properties; microstructural analysis, EDAX

There are too many keywords in the list of keywords. I suggest revising the list of keywords to “Keywords: AISI 316L; heat treatment; tungsten inert gas welding; selective laser melting; mechanical properties; microstructural analysis”.

Page 1: “For example, the Chen team made a successful attempt,…

Please revise “the Chen team” to “Chen’s team”.

Page 2: “However, apart from the process shortcomings mentioned, additive technology (AM) offers advantages such as weight optimization,…

Please revise “additive technology (AM) ” to “additive manufacturing (AM) technology”.

Page 2: “In terms of the final product, additive manufacturing (AM) is limited by the size of the object dimensions.

Please revise this sentence to “In terms of the final product, AM is limited by the size of the object dimensions.

Page 2: “The input material used was 316L austenitic stainless-steel powder with an average particle size between 15 μm and 45 μm [18].

Please revise this sentence to “The materials used for the SLM was 316L austenitic stainless-steel powders with particle size ranging from 15 to 45 μm [18].

Page 2: “Therefore, the powder can be reused successfully, as demonstrated by studies [21, 22].

Please revise this sentence to “Therefore, the powder can be reused successfully, as demonstrated by previous studies [21, 22].

Page 2: “The chemical composition of the powder used is shown in Table 1, which was determined by optical emission spectrometry (GDOES) using a Spectrum Analytic optical emission spectrometer in accord-80 ance with the work [23, 24].

Please revise “optical emission spectrometry (GDOES)” to “glow discharge optical emission spectrometry (GDOES)”.

Page 2: “Table 1. Chemical composition of AISI 316L base material in Wt.%

Please revise “Wt.%” to “wt.%”. Furthermore, balanced of Fe should also be included in the table.

Page 2: “All samples were printed in one structure with identical process parameters; see Table 2.

The dimension of the as-built materials should be indicated in the manuscript.

Page 3: “Figure 1. Test plates made by the SLM method ready for welding

A scale bar should be added into the figure.

Page 3: “In the first stage of heat treatment, heating was carried out at a rate of 250 ° C per hour to an annealing temperature of 1040 °C, with an annealing time of 0,5 h and 2 hours.

Please describe why 1040 °C was chosen as the temperature for heat treatment.

Page 3: “The TIG method (141) was used to weld all test specimens.

What does “(141)” mean?

Page 4: “Figure 2. Schematic of the experimental welded joint arrangement

The unit of this figure is missing.

Page 4: “Figure 3. Location of test specimens for tensile test on welding plate

The unit of this figure is missing.

Page 4: “The samples were tested on a Zwick / Roell Z250 machine or Zwick/Roell Z600 (Zwick Roell Group, Ulm, Germany).

Testing parameters of the tensile test such as strain rate should be indicated in the sentence.

Page 4: “In each case, 19 punctures were made across the entire welded joint and base material.

Please revise “punctures” to “indentations”.

Page 5: “Figure 4. Layout of indenters during the hardness test Vickers HV10

The location of base metal, HAZ and weld metal should be indicated in the figure.

Page 5: “2.6 SEM Analysis

The abbreviation “SEM” should be spelled out during first appearance in the main text.

Page 5: “Figure 5. Tensile test results in the base material horizontal direction of specimens

Please narrow the range of the Y-axis.

Page 6: “Figure 6. Tensile test results in the base material, vertical direction of specimens

Please narrow the range of the Y-axis.

Page 7: “Figure 9. Interval plot of UTS of transverse tensile test – welded joint

The abbreviation “UTS” should be spelled out during first appearance in the main text.

Page 7: “The figure shows that the weld joint without heat treatment exhibits a higher hardness than the annealed specimens, but only in the region of 197 the base material.

There is no discussion related to the hardness values in the WM and in the HAZ.

Page 7: “Figure 10. Comparison of hardness in welded joint

The unit of the X-axis is missing.

Page 7: “For microstructural analysis, the samples were prepared in a standard way, i.e., by grinding, polishing and subsequent electrochemical etching. An Olympus GX- 51 metalographic microscope (Tokio, Japan) was used to observe the microstructures.

Details of the microstructural characterization method should be included in Section 2 Experimental Methods.

Page 8: “The typical structure is meant to be 'weld beads'; see Figure 11a.

Please define the term “weld beads” in the manuscript.

Page 8: “Heat treatment (Fig. 12b) caused only a coarsening of the austenitic grain.

The grain coarsening is not obvious based on the results shown in Fig. 12.

Page 9: “For better understanding was performed EDAX semiquantitative chemical composition analysis in the place of “black particles”.

The abbreviation “EDAX” should be spelled out during first appearance in the main text.

Page 9: “The results of this analysis are shown in Figure 16 and Table 4. The chemical composition corresponds to the composition of the base material - steel 316L. Thus, they are most likely pores, not precipitates. We obtained the same results for all measurements on these “black particles”.

The carbon content of the “black particles” are 3.81 wt.%, which is much higher than the matrix (0.016 wt.%). Are you sure that these “black particles” are not chromium carbide?

Reviewer 2 Report

The article is about heat treatment of steel AISI316L produced by the SLM method. However, some issues must to be addressed:

  1. Abstract: Please define or try to avoid using abbreviations in the abstract. Typically, the abstract should provide a broad overview of the entire project, summarize the results, and present the implications of the research or what it adds to its field.
  2. Figure 1 is useless without providing the sizes of the samples.
  3. Table 3 is useless: replace it with heat treatment diagram !!
  4. Line 110: (141)???
  5. Figure 2: what kind of the joint was prepared: butt joint, tee joint, corner joint, lap joint, edge joint or else?!
  6. Because of lack of clear explanation, figure 3 has no scientific meanings …
  7. Hardness test section: please take into consideration to emphasize where were carry out the investigations: on top, bottom, lateral of the welded joint …??!?
  8. Figures 5-7, 9: please scale the graphs.
  9. Optical microscope is missing in section 2 …
  10. The results are merely presented, not properly discussed. Please add explanations for the observed changes. Please give an extended discussion on the obtained results and correlate your findings with previous literature studies and prospective applications. 
  1. The authors must to provide some details about importance of the research and their applicability.
  2. Please enhance the clarity of the conclusion section in order to highlight the results obtained.
  3. General check-up and correction of the English language is suggested. There are still some minor typos and grammatical errors.

The author needs to address the abovementioned points for the betterment of the manuscript.

Reviewer 3 Report

I would like to acknowledge authors for a very interesting manuscript “Influence of Heat Treatment of Steel AISI316L Produced by the SLM Method on the Properties of Welded Joint”. Unfortunately, I cannot recommend this manuscript for publication without major revision.

Major comments:

1) Never use abbreviations in the title, e.g. SLM. Abbreviations can be used only for standard steel types etc. “SLM” can have many different meanings.

2) The manuscript contains a lot of repetitions. Results of the manuscript can be published as a short letter rather than a standard manuscript.

3) Description of figures 5-9 is ambiguous: the caption should explain the figure without referring to the text. What is UTS-Rm? Why is it represented as a function of time? What are green, red and blue lines? If authors have only tree experimental dots, what kind of assumptions allowed them to draw the straight lines through those dots? This should be clear from the caption of the figure.

4) Can figures 5-9 be combined into two figures only?

5) Figures 11-13 can be combined into one figure.

6) Section “Discussion” looks more like the summary of the section “Results” and the corresponding text would be more suitable for the section “Conclusions”.

Minor comments:

1) page 2 line 64: “heat treatment is carried out in different variations.” Difficult to understand what are “different variations”? Different sequence of thermal annealing steps? Could authors check English language?

2) A lot of paragraphs start with “Figure shows...”. As a result, the description looks like an undergraduate report.

Round 2

Reviewer 1 Report

Very good revision. The authors have successfully addressed the reviewer’s comments. The revised manuscript has meet the requirement for publication.

Reviewer 2 Report

The article was improved and now it is suitable for publication.

Reviewer 3 Report

I still consider the content of this paper to be more appropriate for a short letter and not for a full manuscript: there are not too much results for the full manuscript.